# Zero-shot capability of 2D SAM-family models for bone segmentation in CT scans

**Caroline Magg** [1,2]                C.MAGG@AMSTERDAMUMC.NL
**Clara I. Sánchez** [1,2]
**Hoel Kervadec** [1,2]

[1] *University of Amsterdam, The Netherlands*

[2] *Amsterdam UMC location University of Amsterdam, The Netherlands*

**Editors:** Accepted for publication at MIDL 2025

## Abstract

The Segment Anything Model (SAM) and similar models build a family of promptable foundation models (FMs) for image and video segmentation. The object of interest is identified using prompts—user provided input such as bounding boxes or points—and the models have shown very promising results when it comes to generalization to new tasks. However, extensive evaluation studies are required for medical applications, to assess their strengths and weaknesses in clinical settings. As the performance of those models is highly dependent on the quality and quantity of their prompts, it is necessary to thoroughly benchmark the different options. Currently, no dedicated evaluation studies exist specifically for bone segmentation in CT scans. Leveraging high-quality private and public datasets on four skeletal regions, we test the zero-shot capabilities of SAM-family models for bone CT segmentation, using non-interactive prompting strategies, composed of bounding box, points and combinations of the two. Additionally, we design a guideline for informed decision-making in 2D non-interactive prompting based on our insights on segmentation performance and inference time. Our results show that *SAM* and *SAM2* currently outperform medically fine-tuned FMs, and prompted with a bounding box together with a center point have the best performance across all tested settings. Our code is available in this github repository.

**Keywords:** Segment anything model, Medical image segmentation, Foundation models, Bone segmentation

## 1. Introduction

The release of Segment Anything Model (*SAM*) (Kirillov et al., 2023) started a family of promptable foundation models (FMs) for segmentation. Spatial information in form of bounding box and points inside and outside the object are used as prompts to identify the object of interest. FMs are trained on huge datasets (i.e., hundreds of thousands of images and masks) and their design allows them to generalize to unseen tasks and data. As data scarcity and domain shifts are common problems in medical image segmentation, FMs appear as an alternative to fully supervised, specialized models trained on annotated data.

Since *SAM* and *SAM2* are trained on natural image materials, there remains a gap in applicability for medical data due to the modality differences (natural images vs. medical scans) and image size (2D vs. 3D, at much higher resolutions). Efforts to address this gap have focused on fine-tuning and modifying the *SAM*-architecture to improve its suitability for medical imaging, resulting in versions such as *Med-SAM* (Ma et al., 2024), *SAM-Med2d*

(Cheng et al., 2023b), *Sam-Med3d* (Wang et al., 2024), *Med-Sam2* (Zhu et al., 2024). Beyond model adaptations, thorough evaluation studies are essential to understand the current performance behavior, to identify potential weaknesses, risks and limitations in clinical settings and to formulate application guidelines for medical use cases.

Table 1 shows evaluation studies closely related to our work, especially those conducted with a variety of medical image datasets. The conclusion of several evaluation studies (Mazurowski et al., 2023; He et al., 2023; Mattjie et al., 2023; Cheng et al., 2023a; Huang et al., 2024; Dong et al., 2024) is that performances are unstable across different datasets and task. The models tend to struggle with small, irregular structures with low-contrast or fuzzy boundaries, leading to unsatisfying results. In contrast, they show promising results on larger structures with clear, sharp boundaries. Given that bone appears in CT scans with high-intensity values and well-defined boundaries, we hypothesize that Sam-family models are well-suited to achieve promising results for this task. However, there is no dedicated study focused on CT scans for bone segmentation. In addition, existing studies primarily evaluate the performance of only Sam and Sam2 with a very limited set of prompting options. As the model prediction is directly driven by the provided prompts, it is essential to evaluate a broader variety of options (e.g., prompt combinations).

The aim of this study is to investigate different non-interactive prompting strategies for Sam-family models on bone segmentation in CT scans under "ideal" 2D conditions, i.e., prompts are based on reference masks without manipulation or human error. We test 9 Sam-family models with up to 32 prompting strategies on four different skeletal regions containing different bone and metal structures. Based on our analysis of segmentation performance and model inference time, we introduce guidelines for choosing a 2D prompting strategy and model considering prompting preferences and inference time constraints.

## 2. Dataset

Medical SAM versions (e.g., *Med-Sam*, *Sam-Med2D*, or *Sam-Med3d*) are fine-tuned on publicly available datasets containing bone segmentation from CT scans (e.g., TotalSegmentator (Wasserthal et al., 2023), CTPelvic1K (Liu et al., 2021), VerSe2020 (Sekuboyina et al., 2021)). Therefore, private datasets are required for a fair and independent evaluation across all models, while public datasets enable other researchers to reproduce findings. To achieve this balance, we compiled a private dataset from the department of Orthopedic Surgery and Sports Medicine of the Amsterdam UMC of 80 CT scans from three skeletal regions. Additionally, we selected the TotalSegmentator dataset (Wasserthal et al., 2023) as the public dataset for comparison (D4), as it includes a pre-defined train-and-test split. Although neither *Med-Sam* nor *Sam-Med2D* specify their exact dataset splits, all models in our work are evaluated on a subset of the test set[1]. Thus, our dataset consists of 80 private and 71 public CT scans of four different skeletal regions, annotated with labels for various bone structures and one metal structure (Figure 1). Extra dataset details can be found in Appendix A. The public samples are only used for the comparison to the private dataset, the remaining evaluation is performed on the private dataset alone.

---

1. https://zenodo.org/records/10047292, dataset v2.01

Table 1: Overview of evaluation studies similar to our work. Unknown model sizes are denoted as N/A. † corresponds to prompt application for the largest and multiple disconnected components of one object. The last column indicates whether bone CT segmentation is included in the dataset. ✗* corresponds to testing on X-Rays of the hip (Gut, 2021).

| Reference | Prompting strategies | Models | Dataset | Bone CT |
|---|---|---|---|---|
| (Roy et al., 2023) | • pos. random points 
 • perturbed bounding box | SAM (size N/A) | 1 public | ✗ |
| (He et al., 2023) | • mass center 
 • dilated bounding box | SAM (size N/A) | 12 public | ✗ |
| (Mazurowski et al., 2023) | • center† • bounding box† 
 • simulated interactive 2D prompting | SAM (size N/A) | 19 public | ✗* |
| (Huang et al., 2024) | • center of mass/positive point 
 • 5 pos./pos. and neg. random points 
 • bounding box 
 • bounding box + 1 positive point | SAM (all sizes) | 53 public | ✓ |
| (Cheng et al., 2023a) | • bounding box 
 • center point of bounding box | SAM (size N/A) | 12 public | ✗ |
| (Mattjie et al., 2023) | • center 
 • (distributed) pos. random points 
 • (perturbed) bounding box | SAM (all sizes) | 6 public | ✗* |
| (Dong et al., 2024) | • center† 
 • bounding box† | SAM2 (size N/A) | 18 public | ✗* |
| (Shen et al., 2024) | • pos. and neg. points 
 • bounding box | SAM, SAM2, Med-SAM, SAM-Med2 (sizes N/A) | 1 public | ✗ |
| (Sengupta et al., 2024) | • pos. random points 
 • 1 pos. and 2 neg. points | SAM & SAM2 (all sizes) | 11 public | ✗ |
| (Yu et al., 2024) | • 1 point (undefined) 
 • bounding box | SAM & SAM2 (size N/A) | 2 public | ✗ |
| Our study | • center† • centroid† • bounding box† 
 • pos random points† 
 • bounding box + center† 
 • bounding box + pos/neg. points† 
 • center + 1,5 neg. points† 
 • 1,5 pos. + neg. points† | SAM, SAM2, Med-SAM, SAM-Med2d (all sizes) | 3 private + 1 public | ✓ |

## 3. Methods

### 3.1. SAM-family

**SAM** The Segment Anything Model (*SAM*) (Kirillov et al., 2023) was introduced as promptable "foundation model for image segmentation". *SAM* supports sparse prompts, i.e., bounding box and points (positive and negative), and dense prompts, i.e., masks. The architecture consists of three parts: First, the image encoder, a Masked Autoencoder (MAE) pre-trained Vision Transformer (ViT), is run once per image to create image embeddings of the 2D image input. Then, the prompt encoder creates prompt embeddings for each prompt type. Finally, the lightweight mask decoder combines both embeddings and an output token and predicts the final segmentation mask. The model is available in three different sizes: base (B), large (L) and huge (H), which depends on the ViT encoder.

**SAM2** (Ravi et al., 2024) is an extension of *SAM* with the additional capability of video segmentation. This is realized by changes in the architecture: The ViT encoder is replaced

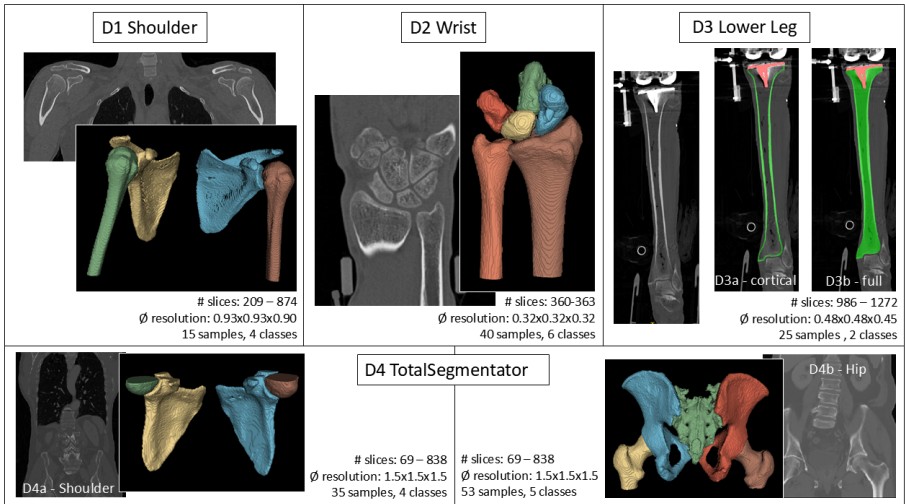

Figure 1: Dataset composition: private dataset containing 80 CT scans from three skeletal regions: shoulder (D1), wrist (D2) and lower leg (two sets of tibia segmentation, cortical (D3a) and full (D3b)); and public dataset containing 71 CT scans from two skeletal regions: wrist (D4a) and hip (D4b).

by a MAE pre-trained Hiera image encoder, and a memory mechanics is introduced to fuse frame embeddings with past frame features and predictions. Due to different Hiera sizes, four different versions are available: base plus (B+), tiny (T), small (S) and large (L).

**Med-SAM** (Ma et al., 2024) was introduced as "a foundation model for promptable medical image segmentation". Without any adaptions to the Sam architecture, *Sam B* is fine-tuned on a medical image dataset with focus on cancer types. *Med-Sam* only supports bounding boxes as it was only retrained for this prompt type.

**SAM-Med2d** (Cheng et al., 2023b) was developed by fine-tuning *Sam B* on SA-Med2D-20M (Ye et al., 2023) with an adapter technique using learnable adapter layers. The model keeps the functionality of both sparse prompts, i.e., bounding box and point.

### 3.2. Prompting Strategies

We use non-interactive prompts, which are automatically extracted from the reference masks. A prompt consists of at least one primitive and one component selection criteria.

**Primitives**   There are 5 primitives which are the building blocks for a prompt (Figure 2):

(a) `bounding box`: Tight box enclosing the entire object.

(b) `(EDT) center`: The point the most furthest away from the object boundary (with respect to the Euclidean distance transform). In case of equality, a single candidate is kept randomly. For simplicity, we refer to it simply as `center` from now on.

(c) `centroid`: Center of mass with homogeneous density. Note that there is no guarantee that the centroid is inside the object. Despite this shortcoming, we include it for completeness and as other existing work (He et al., 2023) used it.

(d) `positive` points: Random points within the region. To avoid random points on the border, the reference mask is eroded by a $3 \times 3$ kernel before sampling.

(e) `negative` points: Random points outside the region but close to the border. The mask is dilated in two steps: first with a $5 \times 5$ kernel and then with a $15 \times 15$ kernel. The point(s) is (are) then sampled from the difference between these two dilations.

The prompt primitives are extracted for each component of the reference mask larger than 15 pixels or larger than 5% of the entire component in a slice. Components smaller than the defined criteria are unrealistic to be annotated as bounding boxes are collapsing and not enough points are included to extract 10 diverse random positive points. The thresholds were chosen empirically after dataset inspection.

**Component selection criteria** As shown in Figure 2, anatomical structures can consist of multiple disconnected components in a 2D slice. In our datasets, the number of disconnected components does not exceed 6, which only occurs for less than 10 slices in our entire dataset. Thus, for a prompt, the primitives of either the largest component (*1C*, denoted as open symbols) or up to 5 components (*5C*, denoted as closed symbols) are used.

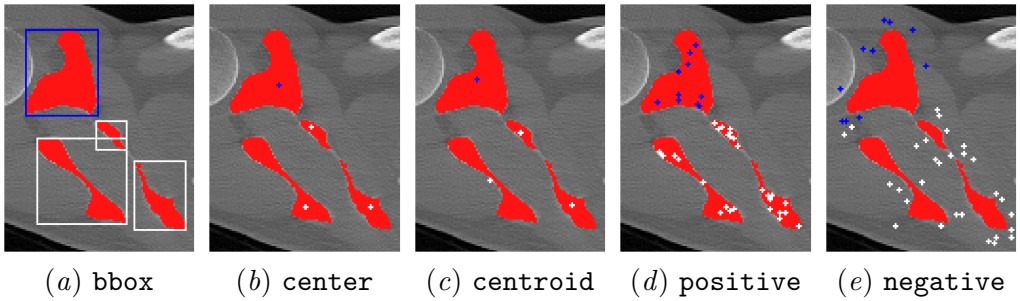

(*a*) `bbox`  (*b*) `center`  (*c*) `centroid`  (*d*) `positive`  (*e*) `negative`

Figure 2: 5 prompt primitives form the building blocks. The largest component's prompt is blue (i.e., one component, *1C*)), while the others are white, resulting in the strategy with multiple components (*5C*), when all components are used.

**A 2D Prompting strategy** (also referred to as "prompt") is defined by one or more prompt primitives and one component selection criteria. They can be divided into three categories based on the primitive types:

- One-type prompts (OT prompts): bounding box ( □ ), center ( ⊙ ), centroid ( ○ ), 1, 3, 5 or 10 positive random points ( △ , ▷ , ▽ , ◁ ).

- bounding box + point combination prompts (BPC prompts): bounding box with center ( ▣ ), with 1 or 5 positive random points ( ⊞ , ◈ ), with 1 or 5 negative random points ( ⊠ , ◈ ).

- Point based combination prompts (PB prompts): center with 1 or 5 negative points ( ⋀ , � ), 1 or 5 positive and negative random points ( ⋀ , ⋁ ).

The centroid is an unreliable primitive since it may lie outside the object depending on its shape, so it is not used for any combinations. Random points combinations are evaluated

with one and five points to compare against the center point and evaluate the impact of increased number of points. In total, there are 32 prompting strategies per model, with exception of *Med-SAM*, which only supports bounding boxes. Since the evaluation on D4 serves as a secondary objective to compare with D1 and expand the number of skeletal regions, the prompting strategies are restricted to bounding box ( □ ), center ( ⊙ ), and bounding box with center ( ▣ ). We refer to the combination of a SAM-family model prompted with a specific prompting strategy as "setting".

### 3.3. Guidelines

We derive guidelines based on two key considerations: First, preference of prompts, influenced by existing workflows or software solutions supporting specific annotations. Second, constraints on inference time and resources, influenced by task-specific requirements (e.g., real-time processing). The guidelines are summarized in a flowchart, with the end-leaves showing the best settings (i.e., highest DSC score on the private dataset) for each condition.

### 3.4. Evaluation

Two common segmentation metrics, Dice Similarity Coefficient (DSC) and 95%-percentile Hausdorff Distance (HD95), were used to compare predictions with reference labels. Moreover, inference time for each model prediction was measured, including the recommended (i.e., no specific bone CT imaging preprocessing with leveling and windowing techniques) image and prompt preprocessing, but excluding image and prompt loading. For multiple prediction calls (as each individual class requires a separate prediction call since binary segmentation masks are returned), the image embedding is done once and reused for all class predictions. As comparison to a fully supervised, dataset-specific model, a 2D and 3D full resolution nnUNet (Isensee et al., 2021) have been trained. Implementation details are available in Appendix C.

## 4. Results

**Segmentation performance**   The segmentation performance of all settings averaged over the private dataset is shown in Figure 3. Considering only segmentation metrics the bottom right corner of Figure 3 shows an overview of the best performing methods with high DSC and low HD95. Visual examples are shown in Figure 4 and Appendix B.3. The best prompting strategy across all models and private datasets is `bbox+center 5C`, which reaches 90.89% DSC and 1.87mm HD95 (Appendix B.1). In comparison, the average 3D full resolution nnUNet performance is 97.74% DSC and 1.72mm HD95, showing a performance gap in favor of nnUNet. Comparing different number of points evaluated on the private dataset demonstrated that the settings with the highest DSC are *SAM H* `10 random points 1C` with 89.6% DSC for a point-based OT setting and *SAM H* `5 positive + negative points 5C` with 91.1% DSC for a PB setting (Appendix B.2).

Analyzing the optimal prompting strategies for each dataset reveals variations across datasets (Appendix B.3). These differences become clear when comparing shoulder CT samples from the public and private data subsets, where, despite similar best DSC scores, the private dataset consistently achieves better DSC across many settings (Appendix B.3.1).

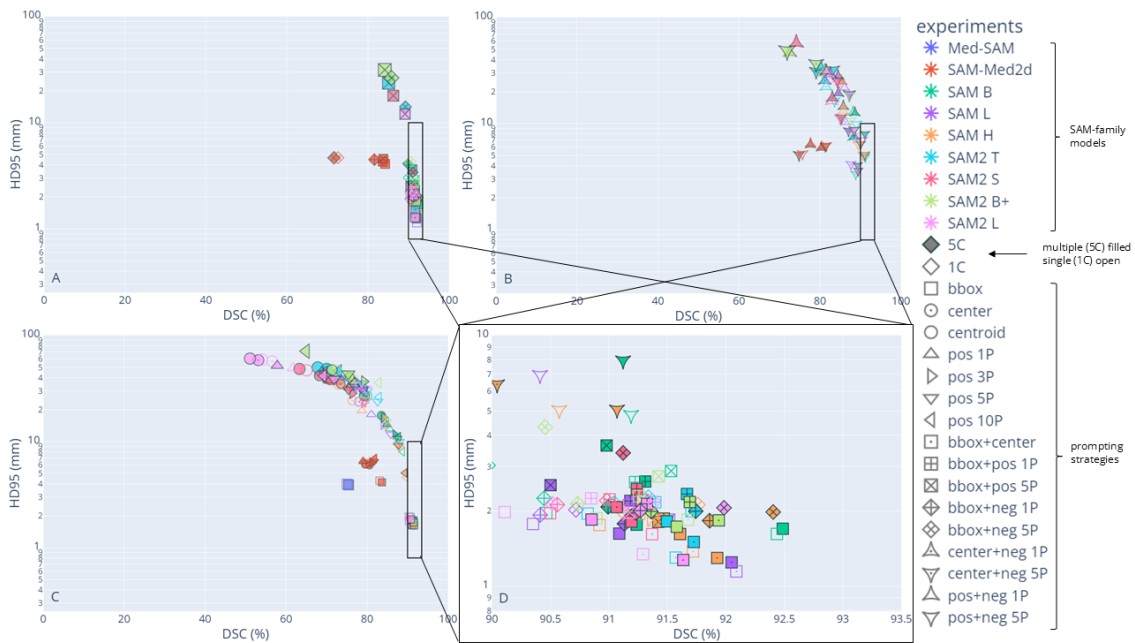

Figure 3: Performance of prompting strategies averaged over private dataset: Scatterplot of (A) BPC prompts, (B) PB prompts, (C) OT prompts, and (D) zoom-in to the lower right corner of subplots (A)-(C). The symbol size in (A)-(C) corresponds to the DSC standard deviation (std), i.e., bigger symbol means higher std.

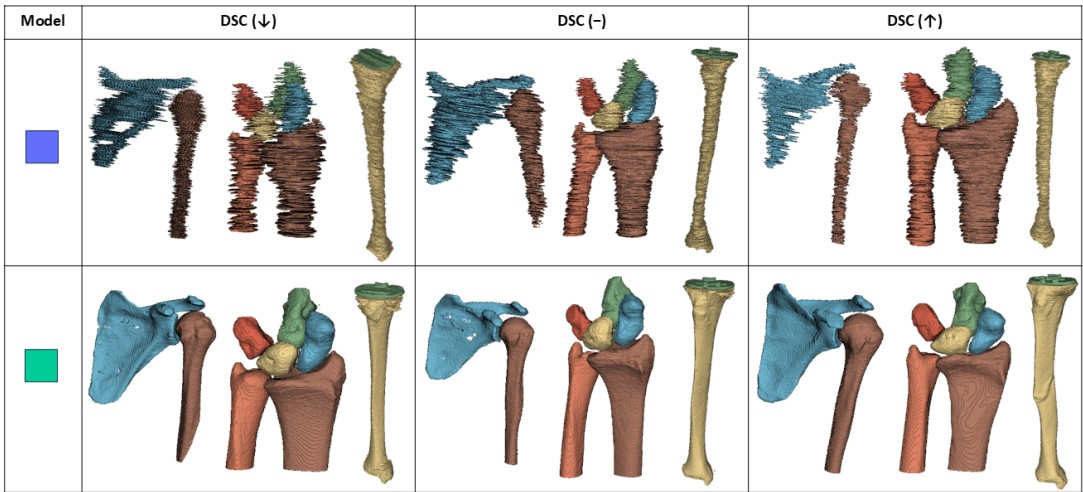

Figure 4: Selected visual examples for `bounding box 5C` predictions for *Med-SAM* ■ and *SAM B* ■ with low (↓), medium (-) and high (↑) DSC (%).

Additional insights come from the lower leg dataset (D3), where different labeling protocols, i.e. cortex versus full tibia bone segmentation, show that the full bone protocol yields superior metrics across all prompting strategies, as cortical bone is significantly over-segmented by the models (Appendix B.3.2).

**Inference time** As inference time per slice (sec.) might be related to number of model parameters, image size and prompting strategies, they are all reported in Table 2. The fastest prediction time has *Sam-Med2D* with 0.054 sec per slice.

Table 2: Average prediction time per slice (sec.): The table on the left sorts the inference time averaged over all prompting strategies in ascending order. The line plot on the right shows the time per slice (sec.) for the different prompting strategies for each model.

| Model | Avg. time per slice (s) | # Model Parameter | Image Size |
|---|---|---|---|
| SAM-Med2d | 0.054 | 271 | 256x256 |
| SAM2 T | 0.068 | 38 | 1024x1024 |
| SAM2 S | 0.080 | 46 | 1024x1024 |
| SAM2 B+ | 0.113 | 80 | 1024x1024 |
| SAM B | 0.166 | 93 | 1024x1024 |
| SAM2 L | 0.240 | 224 | 1024x1024 |
| SAM L | 0.375 | 312 | 1024x1024 |
| SAM H | 0.657 | 641 | 1024x1024 |
| Med-SAM | 1.866 | 93 | 1024x1024 |

**Guidelines** Based on the insights from segmentation performance and inference time, Figure 5 shows our proposed guidelines for non-interactive 2D prompting. Depending on the prompt choice (i.e., no preference, bounding box, combination, one or multiple points) and time restrictions (i.e., low, medium, high), at least one optimal setting is provided.

## 5. Discussion

Based on Figure 3, three trends emerge in segmentation performance of *Sam*-family models. First, performance strongly depends on the prompting strategy. For *Sam* and *Sam2*, their symbols form an arc from optimal to suboptimal metrics, with increasing symbol size (indicating greater DSC standard deviation). Second, *Med-Sam* and *Sam-Med2d*, fine-tuned on medical datasets, are generally outperformed by *Sam* and *Sam2* with most `bbox`-based prompts. Third, zooming into the lower right corner of Figure 3 (D) (high DSC, low HD95) reveals three groups of strategies: `bbox`-only prompts, bounding box + point combinations, and point-based combinations (upper left quadrant). Overall, `bbox+center 5C` achieves the best performance across models and datasets (Figure 6). The evaluation was performed on a shared server, where varying utilization may affect exact inference times. Despite this limitation, clear trends are observed: inference time is primarily influenced by image and model size, not by the prompting strategies (Table 2). *Med-Sam*'s slowest time is potentially due to an inefficient implementation. Based on our results, we propose guidelines for non-interactive 2D prompting that consider both prompt preferences and time constraints (Figure 5). This gives practitioners a much simpler and clearer set of options to pick from, when using FMs on a new applications.

A limitation of our study is that we did not investigate why medically fine-tuned models are outperformed. We suspect this may be due to catastrophic forgetting and loss of general representations, but this requires future testing on the original task (i.e., natural image

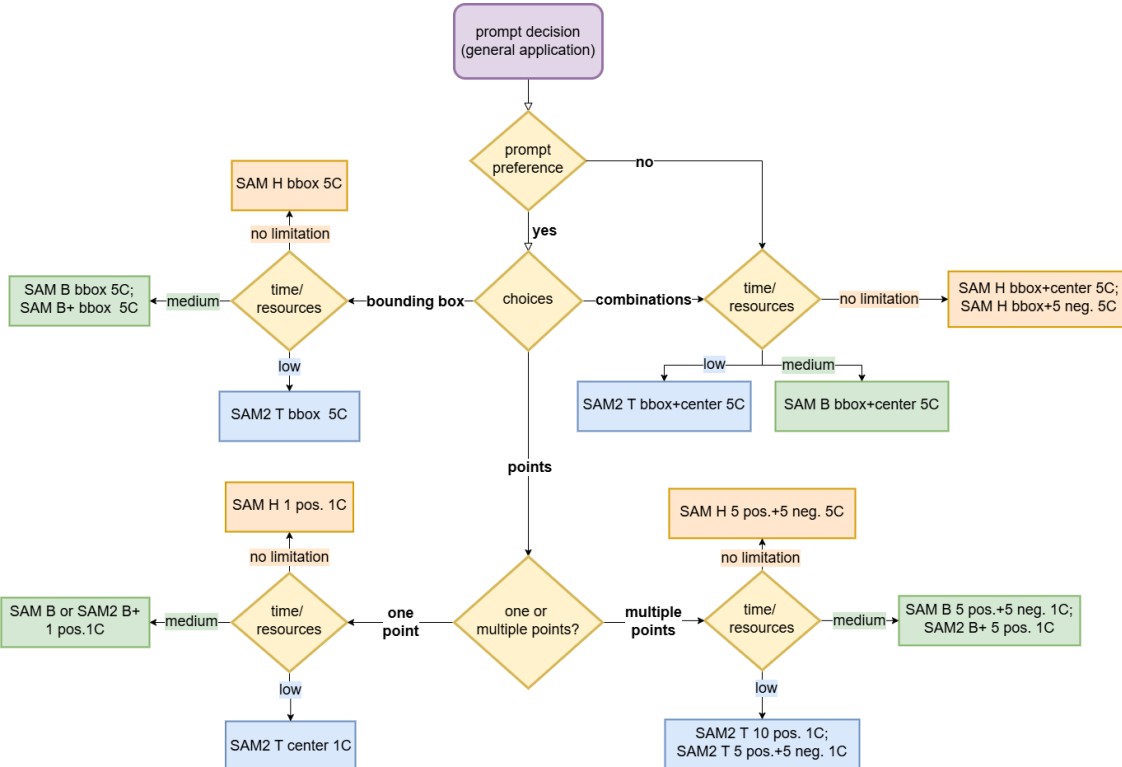

Figure 5: Guidelines for non-interactive 2D prompting for bone segmentation in CT scans based on evaluating 258 settings on 3 private data subsets.

segmentation). Currently, the evaluation and guidelines are limited to "theoretical" conditions, without taking into account human interaction and eventual errors. Although an observer study with human-generated prompts is outside the scope of this paper, based on our current insights, we hypothesize that model performance is robust to small variations in prompt position (e.g., center vs. 1 random point) and influenced by false negative prompts (e.g., 1C vs 5C). In future work we will perform an observer study with multiple observers to confirm these hypotheses. Another future work (intertwined with the observer study) is 3D prompting and 3D models like SAM2, SAM-Med3d, and Med-SAM2. These models offer broader possibilities but also introduce a higher prompting complexity, e.g., slice selection. Insights from our 2D analysis will guide the design of future studies for human-generated prompts and 3D SAM-family models, as it can help limit the prompt primitives to the ones showing better performance in the current study, reducing considerable the amount of experiments and observer time for the analysis.

## 6. Conclusion

We tested 9 different 2D SAM-family models with 32 different non-interactive prompting strategies containing one-type and combination prompts, for bone segmentation in CT scans. Most notably, we found that "vanilla" SAM models consistently outperformed its medically fined-tuned versions. From our results, we derived guidelines for non-interactive 2D prompting to guide practitioners when coming to new applications.

## Acknowledgments

We thank in alphabetical order Leendert Blankevoort, George S. Buijs, Johannes G.G. Dobbe, Arthur J. Kievit, Matthias U. Schafroth, Geert J. Streekstra, Stela Topalova, Lukas P.E. Verweij, and Annemiek ter Wee for their work, support and guidance in data acquisition and curation.

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

## Appendix A. Dataset details

Our dataset consists of 80 private and 71 public CT scans of four different skeleton regions with various different labels for bone structures and one metal structure:

- D1 Shoulder: 15 private bilateral scans with 4 labels for left and right scapula and humerus.

- D2 Wrist: 40 private unilateral scans with 6 labels for capitate, lunate, radius, scaphoid, triquetrum, and ulna.

- D3 Lower Leg: 25 private unilateral scans with 2 labels for tibia bone and tibia implant. There are two different labeling protocols for the tibia bone: cortical bone (D3a) and full bone (D3b).

- D4a Shoulder: 35 public scans with same labels as D1.

- D4b Hip: 53 public scans with 5 labels for sacrum, right and left hip and femur.

All scans of the private dataset were acquired with a Brilliance 64-channel CT Scanner (Philips Healthcare, Best, The Netherlands) or a Siemens SOMATOM Force (Siemens Healthineers, Forchheim, Germany) with 160 mAs, 120 kV. The isotropic voxel spacing is 0.93 mm, 0.32 mm, and 0.48 mm, for D1, D2, and D3, respectively. The annotations were generated using an in-house annotation software (Dobbe et al., 2014) and/or ITK-Snap (Yushkevich et al., 2006).

The TotalSegmentator test set contains 89 scans, of which 18 have been excluded because neither shoulder nor hip labels are present. Of the remaining 71 scans, 16 scans are included in both subsets (D4a and D4b).

## Appendix B. Ablation studies

### B.1. Top 10 prompting strategies

The 10 best performing prompting strategies across different models and datasets are shown in Figure 6. They are determined by ranking the prompting strategy from 1 to 32 (1 being the best) for each model based on their averaged DSC over the private dataset. The best method is `bbox+center 5C`, which reaches 90.89% DSC and 1.87mm HD95 on the private dataset (D1-D3), and 91.24% DSC and 2.98mm HD95 on the public dataset (D4). To compare FMs with fully supervised and task-specific models, 2D and 3D full resolution nnUNets (Isensee et al., 2021) were trained for each private data subset. The training details are reported in Appendix C.1.

### B.2. Different number of points

In Figure 7, prompting strategies with different number of points evaluated on the private dataset are compared. For point-based OT prompts, the best DSC is achieved by *SAM H* `10 random points 1C` with 89.6%, followed by *SAM L* `10 random points 1C` with 87.4% and *SAM2 B+* `5 random points 1C` with 87.2%. For PB prompts `5 positive + negative points` performs best for *SAM H* (`5C`, 91.1%), *SAM B* (`1C`, 91.2%) and *SAM L* (`1C`, 90%).

| Prompt | avg Ranking | DSC (%) avg (std) | HD95 (%) avg (std) |
|--------|-------------|-------------------|---------------------|
| ⊡ | 2.38 | 90.89 (10.0) | 1.87 (2.7) |
| ⊡ | 2.88 | 90.80 (10.0) | 1.79 (2.1) |
| ⊠ | 4.38 | 90.49 (10.2) | 2.50 (2.8) |
| ⊞ | 4.50 | 90.44 (9.6) | 2.50 (2.6) |
| ■ | 5.33 | 88.68 (10.4) | 2.25 (2.2) |
| ◈ | 6.25 | 90.15 (9.0) | 2.26 (2.3) |
| ⊞ | 7.38 | 90.22 (9.7) | 2.49 (2.4) |
| □ | 9.67 | 88.03 (10.7) | 2.39 (2.2) |
| ◈ | 10.38 | 87.91 (10.2) | 7.32 (16.1) |
| ⊠ | 10.75 | 87.67 (14.1) | 12.20 (28.1) |

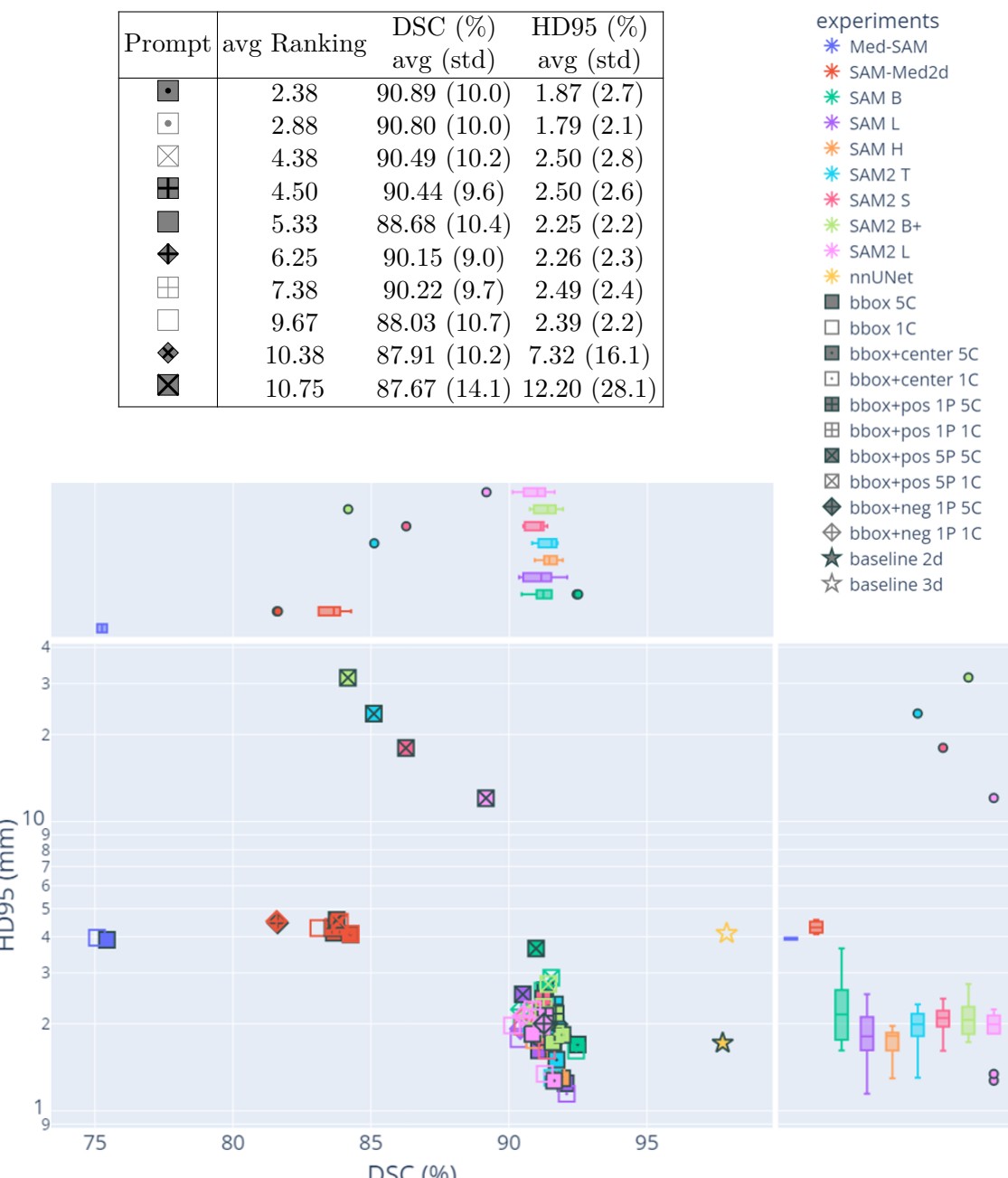

Figure 6: 10 best performing 2D prompting strategies across models. The prompt ranking is determined per model by means of the average DSC over all private data subsets (i.e., highest DSC corresponds to rank 1) and the averaged over all models. The visualization shows the scatter plot with the performance distribution per model across different prompting strategies. Note that the 10 best performing prompts are a subset of the prompts visualized in Figure 3(D).

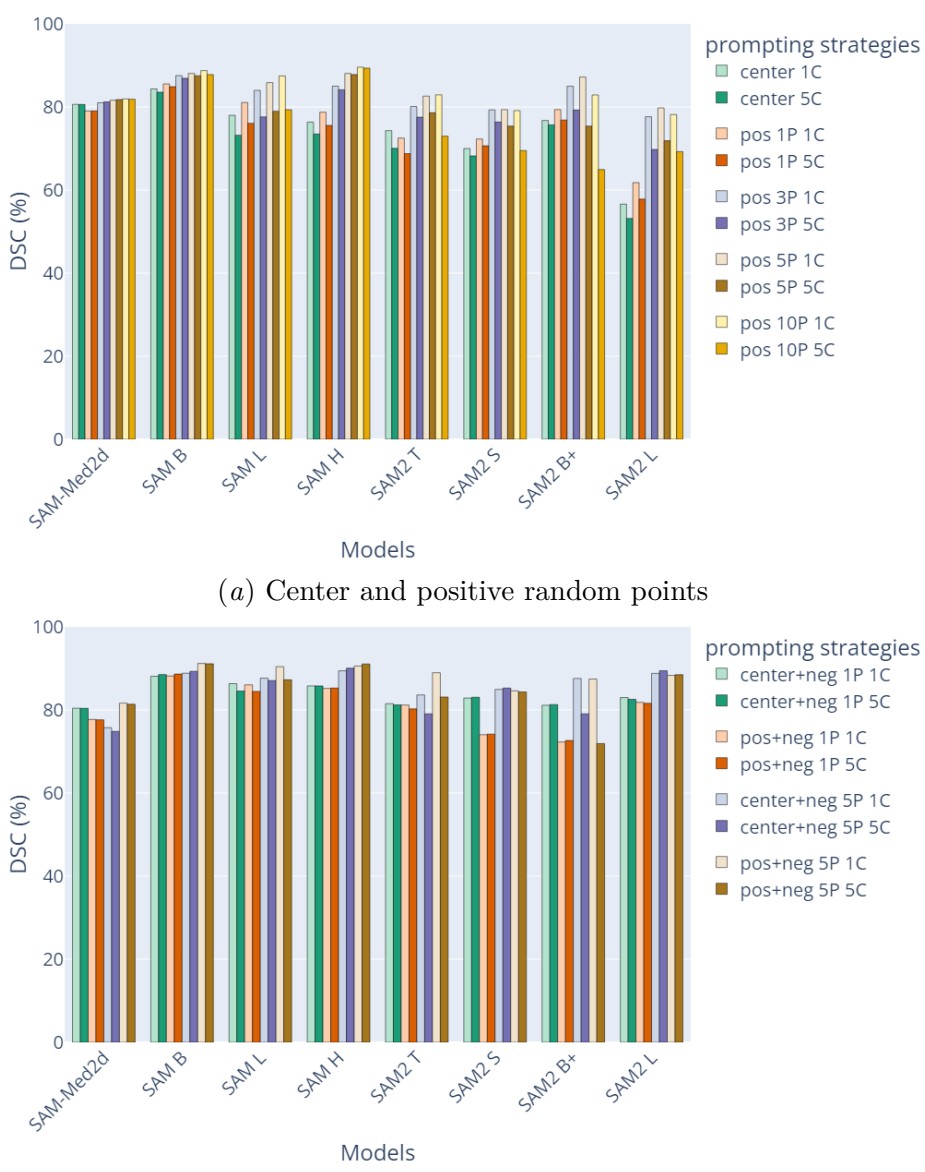

(*a*) Center and positive random points

(*b*) Point combinations

Figure 7: DSC (%) performance for different number of points per model on the private dataset: (a) center point and 1,3,5,10 random positive points; (b) point combinations of center, 1 or 5 positive and negative random points.

## B.3. Dataset-specific analysis

As shown in Figure 8, different dataset have different "best" settings, here i.e., achieving the highest DSC (%). Despite the dataset-specific differences, settings including the bounding box prompt primitive perform the best, with only one exception (SAM performance for D1). A comparison between the best SAM-family setting and a dataset-specific, fully supervised model, such as nnUNet, shows a performance gap in favor of nnUNet.

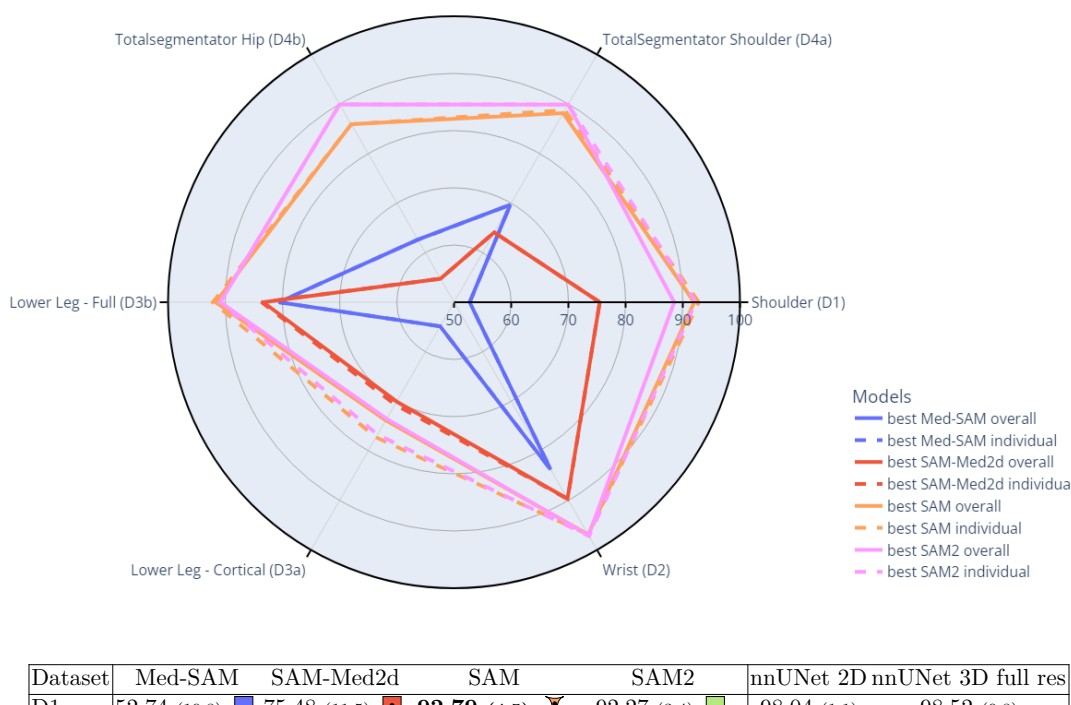

| Dataset | Med-SAM | SAM-Med2d | SAM | SAM2 | nnUNet 2D | nnUNet 3D full res |
|---------|---------|-----------|-----|------|-----------|---------------------|
| D1 | 52.74 (19.8) | 75.48 (11.5) | **92.79 (4.5)** | 92.27 (3.4) | 98.04 (1.1) | 98.52 (0.8) |
| D2 | 83.78 (4.8) | 89.72 (2.4) | 96.91 (0.7) | **97.22 (0.5)** | 98.68 (0.7) | 98.84 (0.4) |
| D3a | 54.86 (29.3) | 70.63 (14.4) | **77.29 (13.5)** | 76.73 (13.3) | 94.95 (3.0) | 93.50 (4.5) |
| D3b | 80.62 (6.7) | 83.66 (3.0) | **92.16 (3.1)** | 90.87 (3.1) | 96.85 (3.0) | 95.82 (4.9) |
| D4a | 62.63 (0.2) | 54.74 (0.2) | 85.93 (0.1) | **89.96 (0.1)** | $-^5$ | 91.48 (5.9)[5] |
| D4b | 69.67 (0.1) | 64.09 (0.1) | 88.78 (0.1) | **89.92 (0.1)** | $-^5$ | 95.11 (2.1)[5] |

Figure 8: Best DSC (%) results per data subset: Radar plot with best setting across subsets (i.e., bounding box for Med-SAM and bounding box + center for remaining models) and with best setting per subset (indicated in table). The table reports the best setting per dataset encoded in the setting symbols and the corresponding DSC (%) scores. The highest scores by a SAM-family model are highlighted in bold for each subset.

---

5. Results from TotalSegmentator v1 for selected labels: https://github.com/wasserth/TotalSegmentator/blob/master/resources/results_all_classes_v1.json, commit 9bd3ca1

B.3.1. Public vs. Private

As we have access to shoulder CT samples with the same label classes from our private dataset (D1) and the TotalSegmentator dataset (D4a), Figure 9 shows the difference in DSC (%) between the two data subsets for a selected subset of settings. For *Med-Sam* and some *Sam2* settings, the DSC on public data is higher, wheras, for *Sam-Med2D* and almost all *Sam* settings, the DSC on private data is higher. As the public dataset was in the fine-tuning dataset of *Sam-Med2D*, the poor results are surprising. However, looking at visual examples (Figure 10) shows that the shoulder joint and humerus are not always fully visible on the CT scans, and, especially center-based prompting settings under-perform on the scapula class, a thin structure with lower contrast.

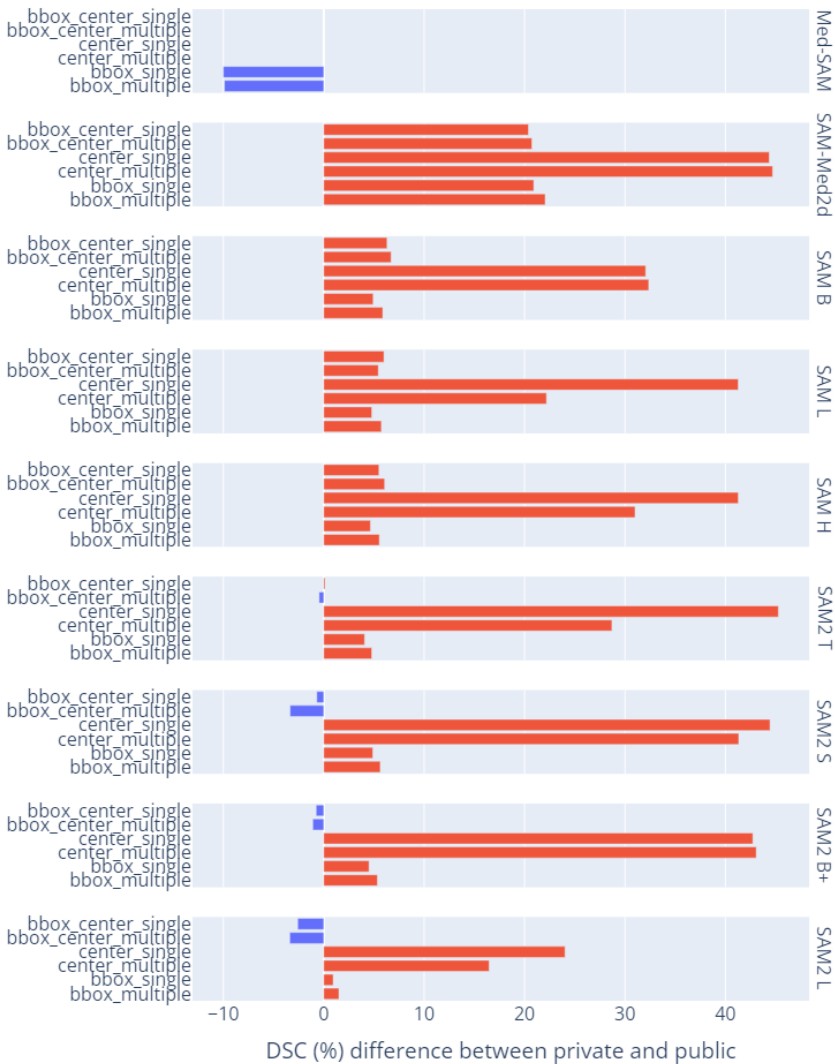

Figure 9: Comparison of private (D1) and public shoulder dataset (D4a) with respect to DSC (%) for selected settings. Red corresponds to private dataset performs better, blue to public dataset performs better.

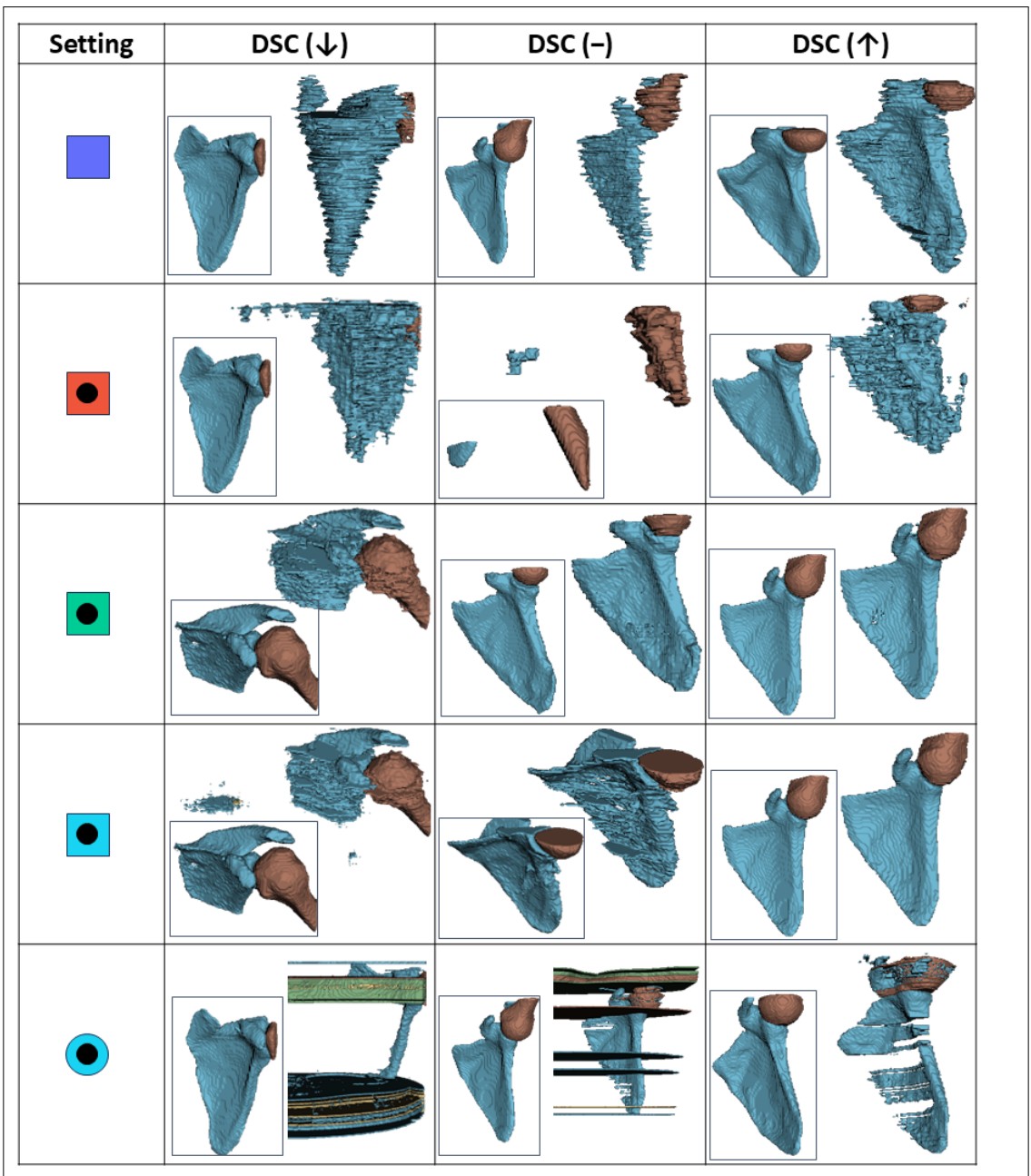

Figure 10: 3D model examples of public shoulder dataset (D4a) for selected predictions (i.e., *Med-SAM* ( 🟦 ), SAM-Med2D ( 🟥 ), SAM B ( 🟩 ), SAM2 B+ ( 🔵 , 🔵 )) and reference labels in the lightgrey box in the lower left corner of each cell with low (↓), medium (-) and high (↑) DSC (%). The labels are color-encoded: blue - scapula, brown - humerus.

### B.3.2. Cortex vs. Full bone segmentation

As shown in Figure 8, a noticeable difference can be seen between the two subsets of the lower leg (D3). The highest dataset-specific DSC scores in both cases are achieved with *Sam B*. However, for D3a, the optimal setting reaches only 77.29% DSC and 5.8mm HD95, while for D3b, the best setting yields 92.16% DSC and 1.9mm HD95. Figure 11 presents the performance of both subsets for *Sam B* across all 2D prompting strategies. The full bone labeling protocol outperforms the cortical protocol, achieving higher DSC and lower HD95 for each prompting strategy. Figure 12 showcases examples of different *Sam B* prompting strategies, which illustrates the over-segmentation in the cortical protocol. A more task-tailored prompting strategy, such as placing negative points in the bone inside (i.e., error regions), might achieve better performance by incorporating dataset-specific knowledge.

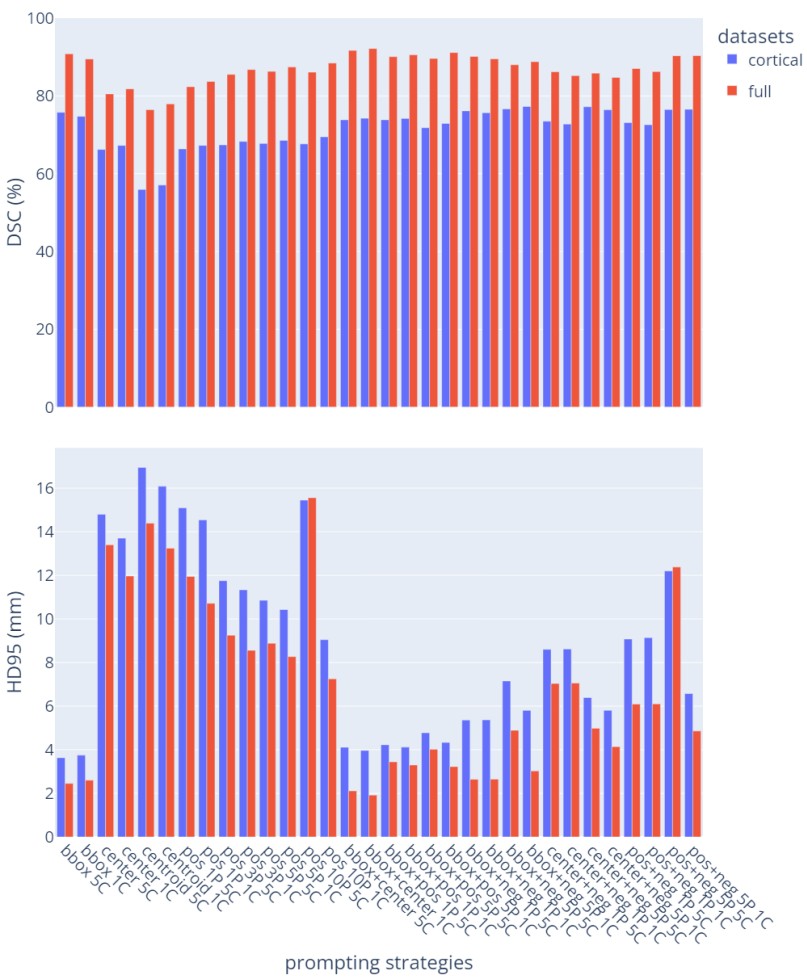

Figure 11: *Sam B* performance (top - DSC (%); bottom - HD95 (mm)) for different labeling protocols in the lower leg dataset (D3), i.e., cortical tibia bone (D3a) and full tibia bone (D3b).

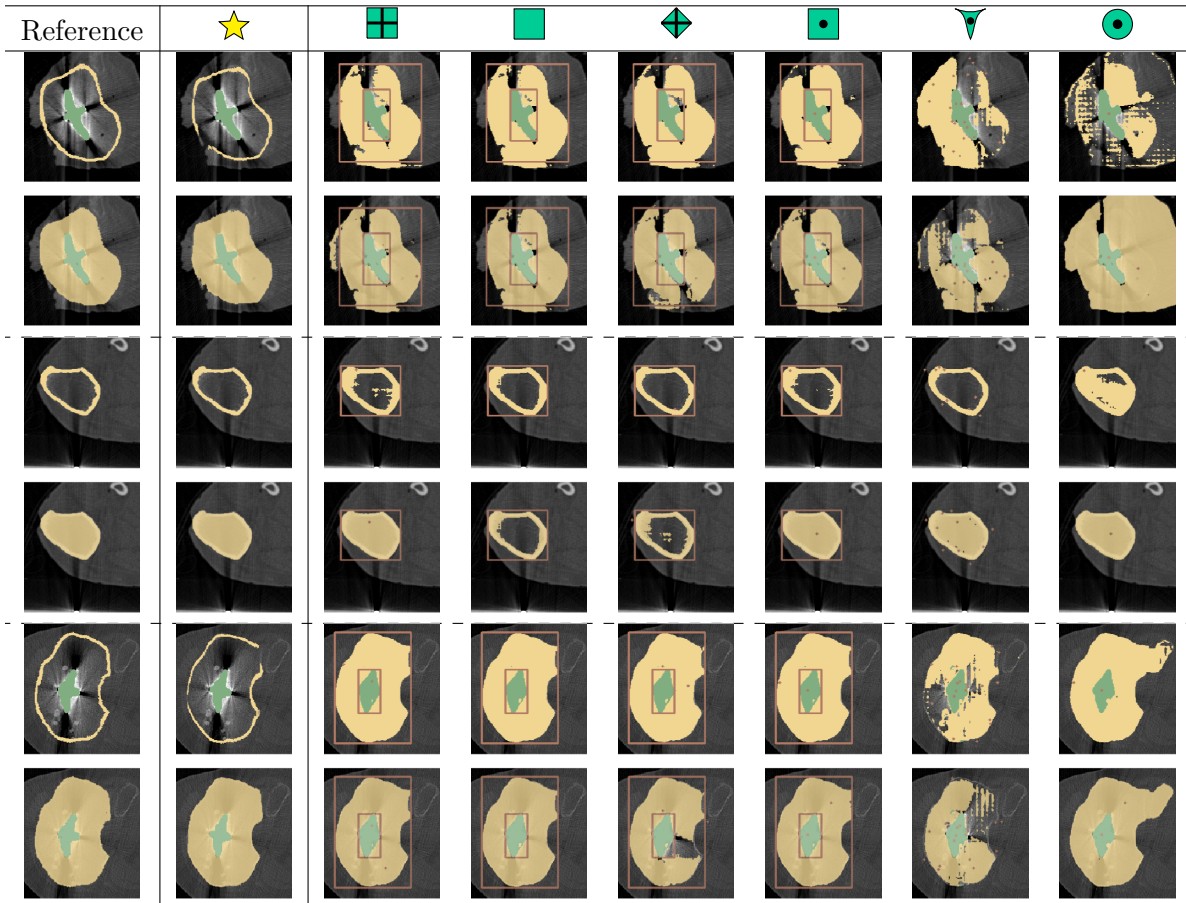

Figure 12: Examples of knee dataset in axial view for reference labels, 2D nnUNet and *SAM B* with different prompt strategies. In each row, the same axial slice is displayed with cortical (top) and full (bottom) tibia bone segmentation. The labels/prompts are color-encoded: yellow - tibia; green - tibia implant; brown - prompts for *SAM B* inference.

## Appendix C. Implementation details

The official github repositories (i.e., *SAM*[2], *SAM2*[3], *Med-SAM*[4], *SAM-Med2d*[5]) are used for all models. Data preprocessing and weight download is performed as instructed. The evaluation is performed on GPUs NVIDIA Geforce RTX 2080 Ti 12GB and an Intel Core Xeon Gold 6128 3.40GHz CPU, which are embedded in a server accessible to multiple users. Evaluation code is adapted from Isensee et al. (2021) and Jia et al. (2024). Visualizations are created with 3D Slicer (https://www.slicer.org/) and plotly (https://plotly.com/).

2. https://github.com/facebookresearch/segment-anything, commit 6fdee8f

3. https://github.com/facebookresearch/sam2, commits 0e78a11 & 29267c8, weights from July 29, 2024

4. https://github.com/bowang-lab/MedSAM, commit 2b7c64c

5. https://github.com/OpenGVLab/SAM-Med2D, commit bfd2b93

## C.1. nnUNet training details

A 2D and a 3D full resolution nnUNet (Isensee et al., 2021) were trained on each of the datasets individually. The default training settings have been retained, except for the data augmentation for D1 and D3 and the division into training and validation folds. For D1, the mirroring on the vertical axes is removed since bilateral scans contain right and left labels. For D3, the mirroring on the horizontal axes is removed since a horizontally flipped femoral bone and implant show some similarity with the tibial counterparts. The models for D1 and D2 are trained and evaluated on a 5-fold, for D3 on a 4-fold patient-based cross-validation split. The results are denoted as ☆ .

