# OpenReview forum: "Zero-shot capability of 2D SAM-family models for bone segmentation in CT scans"
_MIDL.io/2025/Conference — MIDL 2025 Poster_

### Official Review · Reviewer_8aNB · 2025-02-16

**Confidence:** 4
**Preliminary Rating:** 3
**Recommendation:** Poster
**Final Rating:** 4

**Summary:**

This paper presents an empirical study evaluating the performance of various SAM-family model variants in the context of bone segmentation. The authors compare different combinations of non-interactive prompting strategies and report that using a box prompt for each component, in conjunction with a center point and five random positive points, yields the highest-performing prompt configuration. Additionally, the study finds that for bone segmentation task from CT volumes, the three variants of the initial SAM model and SAM2 outperform other subsequent fine-tuning approaches for medical applications.

**Strengths:**

1. The paper provides comprehensive coverage of the various SAM variants and point prompting options, evaluated on both private and public datasets.

2. The visualizations presented in the figures are thorough, well-organized, and aesthetically pleasing.

**Weaknesses:**

1. The focus of this study is limited to bone segmentation in CT images. However, in the conclusion, the authors suggest that their non-interactive 2D prompting strategy could be applied to guide practitioners in new applications. The generalizability of this approach is not guaranteed. Without evidence supporting the broader applicability of their method, the practical utility of this paper may be limited.

2. Hounsfield Units (HU) in CT images carry significant physical meaning. The proper application of leveling and windowing techniques can influence the performance of segmentation models. This important aspect of CT image preprocessing is not addressed in the paper, which could be a potential limitation.

3. Bone segmentation is a well-established task with numerous high-performing, publicly available models, thanks to the high contrast and clear contours of bone, as well as the unique HU range. It is possible to obtain a rough segmentation mask without any fine-tuning using these open-source models. Additionally, both SAM and SAM2 support an alternative prompt style, the "mask" prompt. Expanding the scope of this paper to include and compare the performance of this prompt style would be a valuable addition.

**Detailed Comments:**

N/A

**Justification Of The Final Rating:**

We thank the authors for providing additional clarification regarding the current status of bone segmentation in CT. My concerns about the significance of the study have now been addressed. Given the detailed analysis of prompting strategies for applying SAM-family models to zero-shot bone segmentation in CT scans, I recommend accepting this paper for a poster presentation.

**Justification Of The Preliminary Rating:**

Although the paper offers solid empirical analysis and useful comparisons, its narrow focus, omission of critical factors (such as CT preprocessing), and lack of discussion on broader applicability may restrict the potential impact of the research. Additionally, there is already a significant body of empirical studies on various medical objects in a similar vein. The SAM evaluation in this work lacks substantial novelty or improvements in experimental design, which diminishes the overall value of the contribution.

**Questions To Address In The Rebuttal:**

I encourage the authors to provide a clear justification for the application of their findings, as well as a discussion on the potential implications of their results in broader contexts. This would help clarify the practical significance of their work.

**Special Issue:**

No

---

> ### Author Response · Authors · 2025-03-07
> **Rebuttal for Reviewer 8aNB - Part 1**
>
> > Hounsfield Units (HU) in CT images carry significant physical meaning. The proper application of leveling and windowing techniques can influence the performance of segmentation models. This important aspect of CT image preprocessing is not addressed in the paper, which could be a potential limitation.
>
> "Official" pre-processing instructions provided in the code repositories from the medical fined-tuned SAM were followed as much as possible, which as far as we could tell, does not include leveling and windowing techniques. To the best of our knowledge, neither of the medical fine-tuned SAM versions states specific window width and level values for bone. Therefore, we followed the generic inference code of the official repository, as changing the pre-processing for the application of pre-trained models can decrease the performance, even if it is more modality- or task-specific. This also provided a more consistent pre-processing across all compared models. We have added a small note about the recommended pre-processing in Section 3.4.
>
> > Bone segmentation is a well-established task with numerous high-performing, publicly available models, thanks to the high contrast and clear contours of bone, as well as the unique HU range. It is possible to obtain a rough segmentation mask without any fine-tuning using these open-source models.
>
> As the reviewer commented, considering the high contrast and clear contours of bone, it is expected to publicly available models to perform well. However, our experience so far with these model showed limited usability as well as performance.
> Without explicit reference to a specific public model, we will comment on one commonly used open-source model: TotalSegmentator [1]. The standard license-free TotalSegmentator includes the scapula and humerus classes of our shoulder dataset. The licensed version also includes the tibia, ulna, radius, and one class for all carpal bones in the hand. There is still a gap to the structures that are present in our dataset (tibial implant, differentiation of the carpal bones). Therefore, we cannot apply TotalSegmentator on our entire dataset. On the classes we can apply it on, we get very unsatisfying results, see [Figure 3 and 4](https://www.dropbox.com/scl/fi/ftrfrvomz65cyj5ldkssd/MIDL_2025_examples_rebuttal.pdf?rlkey=y63vuw40kq70fvessiub0wz7s&st=x6mu6jna&dl=0) (link valid until discussion deadline) for examples for the shoulder and lower leg dataset. The predictions for our wrist dataset were empty with respect to the available classes (ulna, radius, carpal).
>
> [1] Wasserthal, J. et al. 2023. TotalSegmentator: Robust Segmentation of 104 Anatomic Structures in CT Images. Radiology: Artificial Intelligence. https://doi.org/10.1148/ryai.230024
>
> > Additionally, both SAM and SAM2 support an alternative prompt style, the "mask" prompt. Expanding the scope of this paper to include and compare the performance of this prompt style would be a valuable addition.
>
> The reviewer is correct, SAM and SAM2 support a mask prompt (dense prompt) in the prompt encoder with 256x256 size. However, early experiments showed that the mask input is not usable as standalone prompt and produced empty results. This has also been discussed by other SAM users in the official repositories ([https://github.com/facebookresearch/segment-anything/issues/169](https://github.com/facebookresearch/segment-anything/issues/169), [https://github.com/facebookresearch/segment-anything/issues/471](https://github.com/facebookresearch/segment-anything/issues/471)). To our understanding, the mask prompt is intended for interactive prompting (which was outside of the scope of our submission), i.e., starting with a point or bounding box prompt and iteratively refining the prompt with the predicted mask as additional prompt. An option would be to investigate the combination of mask and box or point prompt in a non-iterative manner, however, these experiments depend on the initial quality of the masks. Since TotalSegmentator, as shown in the previous comment, does not provide sufficiently good rough masks, we cannot test this prompt combination in the scope of this rebuttal, but will keep it in mind for our future work.
>
> > I encourage the authors to provide a clear justification for the application of their findings, as well as a discussion on the potential implications of their results in broader contexts. This would help clarify the practical significance of their work.
>
> We would like to refer the reviewer to our top-level comment, where we are providing examples on how our findings are already applied. We have also extended the discussion, addressing limitations and how our current work helps future research.

---

> ### Author Response · Authors · 2025-03-07
> **Rebuttal for Reviewer 8aNB - Part 2**
>
> > Additionally, there is already a significant body of empirical studies on various medical objects in a similar vein. The SAM evaluation in this work lacks substantial novelty or improvements in experimental design, which diminishes the overall value of the contribution.
>
> We are aware that there is already a body of evaluation studies on various medical objects, as stated in our introduction. However there is a gap in current validation studies (see provided table below): First, there is no dedicated study focused on CT scans for bone segmentation, for which SAM-family models should be well-suited to achieve promising results. And second, existing validation studies primarily evaluate the performance of either SAM or SAM2 with a limited set of prompting options. In our study, we use 32 different prompting strategies (single prompt types and combinations) with 9 different SAM-family models, which leads to 258 experiments per data-subset.
> To better position our work in the existing body of validation studies, we have added this table to the introduction of our manuscript.
>
> | Authors | Prompting strategies | Models | Dataset | Bone CT |
> |---------|----------------------|--------|---------|---------|
> | Roy et al. (2023) | - pos. random points - perturbed bounding box | SAM (size N/A) | 1 public | ❌ |
> | He et al. (2023) | - mass center  - dilated bounding box | SAM (size N/A) | 12 public | ❌ |
> | Mazurowski et al. (2023) | - center† - bounding box†  - simulated interactive 2D prompting | SAM (size N/A) | 19 public | ❌* |
> | Huang et al. (2024) | - center of mass/positive point  - 5 pos. random points  - 5 pos. and neg. points  - bounding box - bounding box + 1 positive point | SAM (all sizes) | 53 public | ✅ |
> | Cheng et al. (2023) | - bounding box  - center point of bounding box | SAM (size N/A) | 12 public | ❌ |
> | Mattjie et al. (2023) | - center - (distributed) pos. random points  - (perturbed) bounding box | SAM (all sizes) | 6 public | ❌* |
> | Dong et al. (2024) | - center† - bounding box† | SAM2 (size N/A) | 18 public | ❌* |
> | Shen et al. (2024) | - pos. and neg. points - bounding box | SAM, SAM2, Med-SAM, SAM-Med2 (sizes N/A) | 1 public | ❌ |
> | Sengupta et al. (2024) | - pos. random points  - 1 pos. and 2 neg. points | SAM & SAM2 (all sizes) | 11 public | ❌ |
> | Yu et al. (2024) | - 1 point (undefined)  - bounding box | SAM & SAM2 (size N/A) | 2 public | ❌ |
> | **Our study** | - center†  - centroid†  - bounding box†  - pos. random points†  - bounding box + center† - bounding box + pos/neg. points† - center + 1,5 neg. points† - 1,5 pos. + neg. points† | SAM, SAM2, Med-SAM, SAM-Med2d (all sizes) | 3 private, 1 public | ✅ |

---

### Official Review · Reviewer_Zx3j · 2025-02-21

**Confidence:** 5
**Preliminary Rating:** 3
**Recommendation:** Poster
**Final Rating:** 4

**Summary:**

In this paper, the authors investigate the zero-shot performance of the “Segment Anything Model” (SAM) family for bone segmentation in CT scans. They evaluate nine models, both original (e.g., SAM, SAM2) and medically fine-tuned variants (Med-SAM, SAM-Med2D), across four distinct skeletal regions (shoulder, wrist, lower leg, and hip) with 32 different prompting strategies, including bounding boxes, center points, and combinations of multiple prompt primitives. They discover that vanilla SAM often outperforms some medically fine-tuned counterparts, and that combining bounding box and a single center point tends to achieve robust, high-accuracy segmentation. They also provide guidelines for non-interactive 2D prompting to guide practitioners when coming to new applications.

**Strengths:**

1.	The paper lies in its thorough exploration of different prompt types (bounding boxes, centers, positive/negative points, and various combinations).
2.	The authors provide clear, clinically relevant guidelines on how to select a prompt type and a corresponding model based on time constraints and the user’s existing workflow.

**Weaknesses:**

1.	The generalization beyond 2D remains largely unexplored, which limits its applications
2.	Some of the medical variants (e.g., Med-SAM) were shown to underperform, but the specific reasons for this are not deeply analyzed.
3.	The reliance on bounding boxes or center points derived from ground-truth masks does not fully mimic the variability of human annotation, so the validation with real user prompts is needed to strengthen the impact of the findings.

**Detailed Comments:**

1.	See weakness
2.	It would be helpful to clarify how these prompt strategies translate to typical radiologist or surgeon workflows. For instance, do standard PACS tools make it easy to draw bounding boxes or place multiple negative points?
3.	Comparisons to other baseline algorithms (e.g., U-Net variants)
4.	It might be interesting to discuss total time per volume

**Justification Of The Final Rating:**

Most of my concerns have now been addressed, for example, comparisons to other baseline algorithms (e.g., U-Net variants), discuss total time per volume, plans for validating volumetric or fully 3D-based prompts, etc. The authors Therefore, I recommend weak accept.

**Justification Of The Preliminary Rating:**

In this paper, the authors investigate the zero-shot performance of the “Segment Anything Model” (SAM) family for bone segmentation in CT scans. However, due to the weakness mentioned, I recommend Broadline.

**Questions To Address In The Rebuttal:**

1.	Why might certain medically fine-tuned models (e.g., Med-SAM, SAM-Med2D) underperform compared to vanilla SAM?
2.	How does the model handle real-world user prompts with potential inaccuracies?
3.	What are the authors’ plans for validating volumetric or fully 3D-based prompts?

**Special Issue:**

No

---

> ### Author Response · Authors · 2025-03-07
> **Rebuttal for Reviewer Zx3j - Part 1**
>
> > Why might certain medically fine-tuned models (e.g., Med-SAM, SAM-Med2D) underperform compared to vanilla SAM?
>
> Our current hypothesis—supported by years of research observing _castastrophic forgetting_ of deep networks as soon as they are fined-tuned or adapted to another dataset [1, 2]—is that it remains very difficult to fine-tune SAM on a new set of data, without negatively impacting the original generalisation ability [3, 4]. As the original models are trained with 11M samples and 1.1B masks, to our knowledge, the biggest (open-source) medically fine-tuned SAM could collect at best 4.6M samples and 19.7M masks, which is a 2-order of magnitude difference between the two. This is likely an under-estimation of the discrepancy, as samples for medical variants are splitted between quite different modalities, whereas natural images do not have this issue.
>
> At this point, we cannot investigate whether our hypothesis about catastrophic forgetting is true, as this would go beyond the scope of the rebuttal. We have, though, added this to the discussion as a limitation of our findings.
>
> [1] Kemker et al., Measuring Catastrophic Forgetting in Neural Networks, AAAI 2018
> [2] De Lange, Matthias, et al., A continual learning survey: Defying forgetting in classification tasks, TPAMI 2021
> [3] Wang et al., SAM-Med3D-MoE: Towards a Non-Forgetting Segment Anything Model via Mixture of Experts for 3D Medical Image Segmentation, MICCAI 2024
> [4] Zhang, Yuhui, et al., Why are visually-grounded language models bad at image classification?, arXiv preprint  2024
>
> > How does the model handle real-world user prompts with potential inaccuracies?
>
> While we are waiting to finish the human-reader study to assess realistically what type and how many inaccuracies are present in human-generated prompts (see top-level comment), we can already tentatively answer that question by leveraging our current results. First, the nature of false negatives (missing annotations of disconnected components) is similar to the component selection criteria "1C" vs "5C". This shows that, although SAM-family models may generalize to areas outside of the provided box, the "1C" setting still often shows lower performances (Figure 3, Figure 6). Secondly, by using both center and random points, we already see that the exact position of the point does not drastically change the performance despite some smaller variations (Figure 7). This leads us to believe that human-generated point prompts do not influence the model performance drastically as long as they are inside the object. Thirdly, the most problematic, and difficult type of prompt error, would be false positives, which would be problematic for any machine-learning based method. And finally, from other related work mentioned in the top-level comment, preliminary results show that bounding boxes that are too loosely around the object lead to oversegmentation, see [Figure 1 and 2](https://www.dropbox.com/scl/fi/ftrfrvomz65cyj5ldkssd/MIDL_2025_examples_rebuttal.pdf?rlkey=y63vuw40kq70fvessiub0wz7s&st=x6mu6jna&dl=0) (link valid until discussion deadline). We assume that the models are robust against small box variations, but performance decreases for too tight or too loose boxes.
> We have extended our discussion to include our hypothesis based on our current insights.
>
> > What are the authors’ plans for validating volumetric or fully 3D-based prompts?
>
> After carefully assessing the possibilities and prompting limitations of each 3D model, we plan on defining a set of 3D prompting primitives similar to our 2D primitives, e.g., the 3D equivalent to the (EDT) center or a 3D bounding box. Considering the current limitations that some available 3D models do not support full 3D prompts, we plan on finding a way to translate 3D prompts into 2D prompts with 3D information. For example, for SAM2, a 3D bounding box can be translated to a single 2D bounding box in one CT slice and the information about the bounding box bounds can be used during propagation, see [Figure 1 and 2](https://www.dropbox.com/scl/fi/ftrfrvomz65cyj5ldkssd/MIDL_2025_examples_rebuttal.pdf?rlkey=y63vuw40kq70fvessiub0wz7s&st=x6mu6jna&dl=0).

---

> ### Author Response · Authors · 2025-03-07
> **Rebuttal for Reviewer Zx3j - Part 2**
>
> > The generalization beyond 2D remains largely unexplored, which limits its applications. The reliance on bounding boxes or center points derived from ground-truth masks does not fully mimic the variability of human annotation, so the validation with real user prompts is needed to strengthen the impact of the findings.
>
> As stated in the top-level comment, we are currently working on an extension of our current work, including 3D models as well as human-generated prompts. Developing such follow-up studies in a cost- and time-efficient manner would not have been feasible without this first work. In these studies we are limiting the prompt types to those ones showing better performance in the current study, reducing considerable the amount of experiments and observer time for the analysis. We, however, have mentioned the lack of this analysis in the current work as a limitation of this study in the discussion, and added how the current study will help for future work.
>
> > It would be helpful to clarify how these prompt strategies translate to typical radiologist or surgeon workflows. For instance, do standard PACS tools make it easy to draw bounding boxes or place multiple negative points?
>
> In addition to the support of simple annotation tools, radiology workstations are usually equipped with additional software and specialized annotation tools that support drawing a bounding box or a point. Thus, an integration of the prompting strategies introduced in our study should be possible.
>
> > Comparisons to other baseline algorithms (e.g., U-Net variants)
>
>  In the following table, we provide the segmentation metrics DSC (%) and HD95 (mm) for additional baseline algorithms trained and evaluated on the same single fold:
>
> | Dataset | nnUNet 2D | nnUNet 3D | 2D ENet | 2D Residual U-Net |
> |---------|-----------|-----------|---------------|-------------------|
> D1 | 97.85%, 0.92mm | 98.34%, 0.92mm | 97.58%, 0.92mm | 98.93%, 0.92mm |
> D2 | 98.58%, 3.90mm | 98.79%, 0.32mm | 84.55%, 5.71mm | 84.82%, 1.01mm|
> D3b | 97.10%, 0.68mm |  96.8%, 33.11mm | 93.60%, 0.82mm | 82.68%, 5.31mm |
>
> Due to the time constraint of the rebuttal period, we are, however, not able to provide results of these algorithms in the same cross-validation manner as we did for nnUNet. Please, note that the hyperparameters of the other baseline models were not tuned. However, these results show that nnUNet serves as a good baseline comparison for a fully supervised, dataset-specific model.
>
> > It might be interesting to discuss total time per volume
>
> Although we did not measure the total time per volume, we can provide an estimation.
> The total time per volume consists of: loading time per volume + $\sum_{c=1}^{C} (\text{loading time for 2D prompt for class c}+ \text{prediction time for 2D slice for class c})*N_c$, with number of classes $C$ (as binary segmentation masks are predicted) and number of prompted slices per class $N_c$ (as not every slice contains every class). To estimate image loading time, we measured the average time to load a sample in one of our private datasets (our samples consist of different slice numbers, as shown in Figure 1). As the prompts are pre-computed and loading from a json file, this loading time should be negligible. As stated in Section 3.5, we measured the inference time for each model prediction including the recommended preprocessing, but excluding the image and prompt loading. As multiple prediction calls per 2D slice are made to cover all classes, the image embedding is done once and reused for all class predictions.  Thus, to simplify our estimations, we will use the number of slices $N$ instead of $N_c$, which will give an upper bound for the total time per volume: loading time per volume + $N$ * average prediction time per slice.
>
> Wrist dataset
> * median slice number = 363
> * average image loading time = 1.64s
>
> | Model      | Time per volume      |
> |------------|---------------------|
> | SAM-Med2D*  | 1.64 + 363 * 0.054 = ~21s |
> | SAM2 T     | 1.64 + 363 * 0.068 = ~26s |
> | SAM2 B+    | 1.64 + 363 * 0.113 = ~43s |
> | SAM B      | 1.64 + 363 * 0.166 = ~62s |
> | SAM H      | 1.64 + 363 * 0.657 = ~240s |
>
> *Note that SAM-Med2D uses 256x256 image size, whereas the others use 1024x1024 image size.

---

### Official Review · Reviewer_E5oE · 2025-02-22

**Confidence:** 4
**Preliminary Rating:** 4
**Recommendation:** Poster
**Final Rating:** 4

**Summary:**

The study tested 9 different 2D SAM-family models with 32 different non-interactive prompting strategies (bounding box, points, and their combinations) on both private and public CT bone datasets. Based on the findings, it establishes guidelines for effective non-interactive 2D prompting in bone segmentation for CT scans.

**Strengths:**

- The authors conducted thorough experiments on diverse 2D SAM-family models with diverse prompting options and presented results in depth.
- The authors used private datasets to ensure that the models have not seen these data.
- The authors established practical guidelines for effective non-interactive 2D prompting in bone segmentation for CT scans, providing valuable insights for the community.

**Weaknesses:**

- The study is limited to testing existing methods and does not introduce any novel methods.
- SAM2 is trained on videos. Therefore, is it possible to develop more advanced non-interactive prompting methods that leverage between-slice relationships in CT scans, similar to how videos utilize between-frame relations? Additionally, recent studies (e.g., [1]) have explored using SAM2 as an in-context learning tool by providing examples, which could potentially outperform point and bounding box prompts. The authors may consider evaluating such approaches as well.

Ref:
1. Zhao, Lin, et al. "Retrieval-augmented Few-shot Medical Image Segmentation with Foundation Models." arXiv preprint arXiv:2408.08813 (2024).

**Detailed Comments:**

- Comparing the results with nnUNet is crucial for readers to gauge the performance of SAM models. Since this comparison is currently only included in the Appendix, the authors may consider highlighting it in the main text.

**Justification Of The Final Rating:**

The manuscript provides a comprehensive evaluation of SAM-like models for medical image segmentation and offers insightful guidelines, , though there is still room for improvement. My rating remains unchanged.

**Justification Of The Preliminary Rating:**

Although the manuscript does not introduce new or innovative methods, it presents a thorough evaluation of SAM models with diverse prompts on CT bone datasets. It offers valuable insights into effectively utilizing SAM with non-interactive prompts for such datasets.

**Questions To Address In The Rebuttal:**

Please see Weaknesses and Detailed Comments.

---

> ### Author Response · Authors · 2025-03-07
> **Rebuttal for Reviewer E5oE**
>
> > Comparing the results with nnUNet is crucial for readers to gauge the performance of SAM models. Since this comparison is currently only included in the Appendix, the authors may consider highlighting it in the main text.
>
> We use the additional page for the camera-ready version and include nnUNet results in the main text, as suggested by the reviewer.
>
> > The study is limited to testing existing methods and does not introduce any novel methods.
>
> The reviewer is correct that we do not introduce novel models. We would like to point out that our submission falls into the validation and application track of MIDL; through our extensive benchmarking of current public models.
>
>
> > SAM2 is trained on videos. Therefore, is it possible to develop more advanced non-interactive prompting methods that leverage between-slice relationships in CT scans, similar to how videos utilize between-frame relations?
>
> SAM2 was trained on natural images and videos, and has some limitations in incorporating between-slice relationships in CT scans. Aside from the 2D prediction mode, there is also a video mode, which can be applied to a whole CT volume by 2D prompting (on one or multiple slices) and propagation.
> A 3D box for a scan naturally translates into a series of 2D boxes for continuous frames in a video. However, this serie of 2D boxes will end-up being very loose around the object (sometimes intolerably so, depending on the task and anatomy), see [Figure 1 and 2](https://www.dropbox.com/scl/fi/ftrfrvomz65cyj5ldkssd/MIDL_2025_examples_rebuttal.pdf?rlkey=y63vuw40kq70fvessiub0wz7s&st=x6mu6jna&dl=0) (link valid until discussion deadline). In addition, we would like to point out that bounding boxes in the video mode are only supported with the introduction of SAM2.1 (introduced 09/30/2024).
> In order to leverage between-slice relationship in volumetric scans, we plan on investigating 3D scribbles for SAM2 and related models and adapt labels such as Extreme Points [1] as prompts. As 3D scribbles introduce multiple degree of freedoms, the evaluation should be carried out carefully.
>
> [1] Dorent, Reuben, et al. "Inter extreme points geodesics for end-to-end weakly supervised image segmentation." Medical Image Computing and Computer Assisted Intervention–MICCAI 2021: 24th International Conference, Strasbourg, France, September 27–October 1, 2021, Proceedings, Part II 24. Springer International Publishing, 2021.
>
> > Additionally, recent studies (e.g., [1]) have explored using SAM2 as an in-context learning tool by providing examples, which could potentially outperform point and bounding box prompts. The authors may consider evaluating such approaches as well.
>
> We thank the reviewer for sharing the mentioned study. As we are focused on zero-shot performance, this goes beyond the scope of our current work. By retrieving similar samples from limited annotated data, the memory mechanism of SAM2 can be utilized for few-shot segmentation. However, there are still a few annotated samples required to build a memory bank. As medical data can show a large variety in anatomy and pathologies, the few examples might not be representative enough and have to be selected very carefully. In order to generate the required examples, SAM-supported annotation tools could prove to be valuable to speed up manual annotation. Our findings, more specifically our guidelines in Figure 5, can support the decision-making of choosing the most effective model and prompting strategy to generate promising pseudo-labels.

---

> > ### Comment · Reviewer_E5oE · 2025-03-12
> >
> > I appreciate the authors' feedback. I would like to clarify that my suggestion to use SAM2 video mode for 3D images refers to utilizing images from previous slices and their corresponding masks by storing them in the memory bank, not simply using the 3D bounding box. That said, the manuscript provides a comprehensive evaluation of SAM-like models for medical image segmentation and offers insightful guidelines. My rating remains unchanged.

---

### Author Response · Authors · 2025-03-07
**General comment to all reviewers**

We thank the reviewers for their valuable time, and their thorough reviews. We are happy that the paper was well received (and even found æsthetically pleasing), and we are keen to engage further during the discussion period.

In this top-level reply we will address some questions shared across reviewers about human-generated prompts, 3D prompting and how our findings are usable/actionable at this stage, before replying to the other points individually.

Prompting strategy, human interaction, iteration frequency and dimensionality (2D/3D) are all aspects that can affect the final performance of zero-shot SAM-family strategies. Our current study focuses on the first variable while keeping the other variables fixed to non-interactive, non-iterative and 2D prompting. This already sheds light in the analysis of SAM-family models for medical images and allowed us to study the segmentation performance and inference time of 32 prompting strategies in 9 SAM-family models, which results in guidelines to support the choice for a SAM-family model and corresponding prompting strategy. We found the results of this paper, since submission, to be already actionable for our own research (see individual points below), and we believe that other researchers in the field can benefit from it as well.

We are currently working, as an extension of this manuscript, on the following:

*  3D prompts and 3D models (not all current models support it, complexifying comparisons and reporting);
*  human-reader study on prompt accuracy;
*  benchmark using human-generated and/or simulated prompts.

Designing these follow-up studies would not have been feasible without this first paper.

Some 3D models do not allow for fully 3D prompting, but resort to 2D prompting with some additional 3D information (e.g., SAM2 video mode is prompted with a 2D prompt and then propagation through the volume is applied). Prompting of 3D models offers greater possibilities but also introduces added complexity, which expands the potential number of prompting strategies. With the current results, we can make informed decisions about which strategies may be excluded for feasibility reasons (e.g., centroid point).

In addition, the current results allow us to design a cost- and time-efficient observer study by selecting the identified optimal prompt types (i.e., bounding box + center) and to escalate the number of participants included in the study (currently aiming for 20-30 participants). Otherwise, we would have been at risk to have a costly study that did not give reliable insights or results. With the focused scope we can study among others:

* human prompts accuracy, compared to the ideal prompts;
* inter-rater variablity;
* segmentation performance with human-generated prompts
* connected-components recall: how often (medically trained) annotators miss some;
* eventual false positives: if extra non-objects get annotated;
* annotation time per sample.

From the observer study, we also hope to be able to estimate the distribution of human accuracy (position of prompts, false true/negatives). This, in turn, would be usable to simulate human prompts based on real-world information, starting from a "perfect" prompt and adding noise sampled from the human distribution, facilitating future benchmark studies.

In parallel to extending the benchmarking, we are also working on deep-networks supervision with 3D bounding boxes (extension of [1], actually), and are using SAM-family as one baseline: with this paper results, we are able to restrict experiments to the most effective and promising SAM versions, reducing unnecessary comparisons and computations.

The very fast pace of development of foundational models makes it difficult at times to benchmark them thoroughly, as shown by the limited number of efforts carried out so far. However, we believe that this first study, currently restricted to 2D prompting, has valuable insights and lessons for the community, while highlighting future interesting areas of research.

---
[1] https://2020.midl.io/papers/kervadec20

---

### Author Rebuttal · Authors · 2025-03-07

**Rebuttal:**

Based on the reviewers' comments and questions, we have made changes to the manuscript (highlighted in red):

* Introduction: presenting related work in a table
* Section 3.4 Evaluation: adding a note about the recommended pre-processing and using nnUNet for comparison to a fully supervised, task-specific model
* Results: adding averaged nnUNet results
* Discussion: adding the limitation of not investigating the reason for the performance gap between medically fine-tuned and original SAM versions, although we suspect the reason is catastrophic forgetting; more information about the planned observer study and how the current findings can help with future work

**Supporting Material:**

/attachment/d61bd21c1af2482894d67d1b2158a8e869e2b7f2.pdf

---

### Comment · Area_Chair_TAGF · 2025-03-09
**Discussion Period**

Dear Reviewers,​

Thanks for your time and effort in reviewing this paper. This is the right time to discuss this paper with each other.​

The authors have provided a rebuttal to your comments and uploaded a revision. Please review their responses and the revised manuscript. For the preliminary recommendation, we have two borderlines and one weak accept.​ Considering the authors' responses and the discussion, please update your rating and assessments for the paper.

Any discussion is welcome, and you may consider reading each other's reviews, posting questions for clarification, and reaching a consensus.​

Best,
Your AC

---

### Comment · Area_Chair_TAGF · 2025-03-14
**Urgent discussion due in about one day**

Dear all the Reviewers,

The discussion period is nearing its conclusion. All three reviewers voted to accept this paper (3 weak accepts). In my view, this paper could be accepted. Any discussion is welcome!

Best,
Your AC

---

### Meta-Review · Area_Chair_TAGF · 2025-03-20

**Recommendation:** Accept (Poster)
**Confidence:** 4

**Metareview:**

The paper explores the zero-shot capabilities of SAM with non-interactive prompting strategies for bone segmentation in CT. The reviewers were fairly positive but raised some questions, in particular about methodological novelty and generalizability. Some of the concerns have been addressed in the revision. Although the scores are not overwhelmingly positive I think that this would make a good conference contribution.